# Big Data Maturity Assessment Models: A Systematic Literature Review

**Zaher Ali Al-Sai** [1,2,*], **Mohd Heikal Husin** [2] , **Sharifah Mashita Syed-Mohamad** [2], **Rosni Abdullah** [2] ,
**Raed Abu Zitar** [3], **Laith Abualigah** [2,4,5,6,7] **and Amir H. Gandomi** [8,9,*]

1. Department of Management Information Systems, Faculty of Business, Al-Zaytoonah University of Jordan, Amman 11733, Jordan
2. School of Computer Sciences, Universiti Sains Malaysia, Penang 11800, Malaysia
3. Sorbonne Center of Artificial Intelligence, Sorbonne University-Abu Dhabi, Abu Dhabi 38044, United Arab Emirates
4. Prince Hussein Bin Abdullah College for Information Technology, Al Al-Bayt University, Mafraq 130040, Jordan
5. Hourani Center for Applied Scientific Research, Al-Ahliyya Amman University, Amman 19328, Jordan
6. Faculty of Information Technology, Middle East University, Amman 11831, Jordan
7. Faculty of Information Technology, Applied Science Private University, Amman 11931, Jordan
8. Faculty of Engineering and Information Technology, University of Technology Sydney, Ultimo, NSW 2007, Australia
9. University Research and Innovation Center (EKIK), Óbuda University, 1034 Budapest, Hungary
* Correspondence: z_alsai@yahoo.com (Z.A.A.-S.); gandomi@uts.edu.au (A.H.G.)

**Abstract:** Big Data and analytics have become essential factors in managing the COVID-19 pandemic. As no company can escape the effects of the pandemic, mature Big Data and analytics practices are essential for successful decision-making insights and keeping pace with a changing and unpredictable marketplace. The ability to be successful in Big Data projects is related to the organization's maturity level. The maturity model is a tool that could be applied to assess the maturity level across specific key dimensions, where the maturity levels indicate an organization's current capabilities and the desirable state. Big Data maturity models (BDMMs) are a new trend with limited publications published as white papers and web materials by practitioners. While most of the related literature might not have covered all of the existing BDMMs, this systematic literature review (SLR) aims to contribute to the body of knowledge and address the limitations in the existing literature about the existing BDMMs, assessment dimensions, and tools. The SLR strategy in this paper was conducted based on guidelines to perform SLR in software engineering by answering three research questions: (1) What are the existing maturity assessment models for Big Data? (2) What are the assessment dimensions for Big Data maturity models? and (3) What are the assessment tools for Big Data maturity models? This SLR covers the available BDMMs written in English and developed by academics and practitioners (2007–2022). By applying a descriptive qualitative content analysis method for the reviewed publications, this SLR identified 15 BDMMs (10 BDMMs by practitioners and 5 BDMMs by academics). Additionally, this paper presents the limitations of existing BDMMs. The findings of this paper could be used as a grounded reference for assessing the maturity of Big Data. Moreover, this paper will provide managers with critical insights to select the BDMM that fits within their organization to support their data-driven decisions. Future work will investigate the Big Data maturity assessment dimensions towards developing a new Big Data maturity model.

**Keywords:** big data; big data analytics; maturity model; capability maturity model (CMM); big data maturity model; COVID-19 pandemic; critical success factors; readiness assessment; systematic literature review



## 1. Introduction

The COVID-19 pandemic disrupted the expectations of the global market and accelerated the digital transformation by roughly five years; no business has escaped being impacted by the pandemic [1–3]. EMC Corporation and Industrial Development Corporation (IDC) announced that the generated data size in 2020 will be greater than 40 zettabytes (ZB). This is more than 44 times the data in 2009 [4,5]. According to the newly updated

report by the Global DataSphere from International Data Corporation (IDC), data of more than 59 zettabytes (ZB) will be captured, consumed, created, and copied during the pandemic. The COVID-19 pandemic is affecting this statistic due to the unforeseen increase in the number of employees working from home and a tangible increase in the utilization of downloaded and streaming videos [6,7].

During the pandemic, regardless of analytics maturity, various organizations developed analytics solutions for faster response [8]. Big Data has a significant impact on supporting decision-making [9]. To keep pace with a changing marketplace, it is more important than ever for your organization to embrace data-driven decision-making. Before organizations can get started, they will need to identify the concepts and insights behind their Big Data and implement an advanced analytics practice to ensure their analytics practice is up to date and set up for successful decision-making insights [10].

Big Data as a critical challenge could be defined as "a term that describes large volumes of high velocity, complex and variable data that require advanced techniques and technologies to enable the process of capturing, storing, distributing, managing, and analyzing the information" [5,6]. Based on the existing studies [5,11–14], there is no unified definition for Big Data between industry and academia. The Statistical Analysis System Institute (SAS) defined Big Data as a "Popular term used to describe the exponential growth, availability, and use of information, both structured and unstructured" [15]. IBM also added a definition for Big Data, "Data is coming from everywhere; sensors that gather climate information, social media posts, digital videos and pictures, purchase transaction record, and GPS signal of mobile phone to name a few" [4,15]. Therefore, Big Data can be considered as both an entity and a process. BD as an entity includes a volume of data captured from various resources (internal and external) and consists of structured, semi-structured, and unstructured data that cannot be processed using traditional databases and software techniques. BD as a process refers to the organization's infrastructure and technologies used to capture, store and analyze various types of data [5,14,16].

Moreover, BD is pointed out as a technology that enables the processing of unstructured data; BD technologies are the systems and tools used to process BD, such as NoSQL databases, the Hadoop Distributed File System, and MapReduce [17,18]. BD provides new insights to discover new values, supporting organizations to benefit from a deep understanding of the hidden values [19]. Big Data analytics (BDA), as technologies (database and data mining tools) and techniques (analytical methods and techniques), can be employed to analyze large-scale and complex data for a variety of applications prepared to increase the performance and effectiveness of the organization [5,14,16].

Despite the new opportunities for organizations to gain faster insights from faster Big Data, the challenges and issues that increase on a large scale also should be handled seriously before and during the implementation [19]. Due to the large volume of generated data, the current state of the organizational structure, technology infrastructure, technology capabilities, processing capacity, and human resources often fails to deal with the high requirements for Big Data [20,21]. Gaining a clear understanding of your company's data maturity is critical to solving many challenges related to Big Data [1].

Big Data projects often differ from related technology projects, as implementing Big Data projects requires new organizational and technical approaches [22]. This often demands that organizations be ready for additional requirements in various areas to address the complexity of the three "Vs" of Big Data characteristics (volume, velocity, and variety) and to increase their ability to gain high-quality value from the Big Data projects [20,21].

The ability to be successful in Big Data is related to the maturity level of the organization [1,5,11,23,24]. "Maturity" is the condition of being ready, complete, or perfect [8,25]. Building an adequate infrastructure that can integrate various sources of variant data can help the organization mature its data analytics capabilities [11]. The required tools for data collection, data warehousing, and reporting technology should be aligned with business needs, objectives, and strategies [26].

The maturity model can be defined as a tool that could be used to evaluate the maturity level regarding particular key dimensions. The organization's present capabilities and desirable state can be represented by the maturity levels [27]. Consequently, the maturity model serves as the scale for evaluating the current state on the transformation path. In addition, the maturity model (MM) could be used to assess the organization's ability to achieve pre-determined goals [27,28]. The predefined activities regarding determining technology resources, infrastructure, and capabilities could successfully guide the organization to implement Big Data analytics [26].

The capability maturity model (CMM), considered the first maturity model, is used to guide the software's development. It was developed in 1986 by the Software Engineering Institute (SEI) [11,25,29–31]. As shown in Figure 1, the CMM was published in 1993 with five (5) continuous maturity stages: 1. Initial; 2. Repeatable; 3. Defined; 4. Managed; and 5. Optimized.

1. Level 1 is "Initial", where processes are not controlled and are unpredictable.
2. Level 2 is "Repeatable", where processes are characterized for specific organizations but are often reactive.
3. Level 3 is "Defined", where processes are standardized and typically documented.
4. Level 4 is "Managed", where processes are measured and controlled.
5. Level 5 is "Optimized", where processes have a focus on continuous improvement.

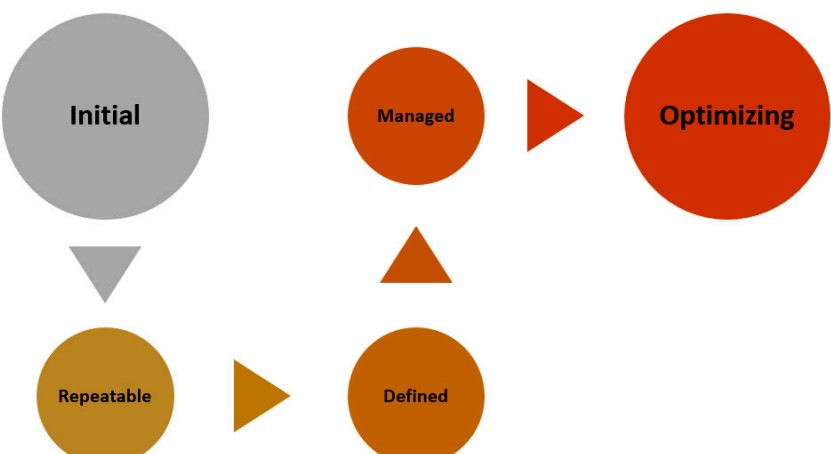

**Figure 1.** Capability maturity model (CMM) [4,5,31].

The CMM was modified to be the capability maturity model integration (CMMI) [11,32,33]. Referring to Figure 2, CMMI was published with five (5) successive stages, namely 1. Initial, 2. Managed, 3. Defined, 4. Quantitatively Managed, and 5. Optimized, to yield an effective improvement in the organization's practices and performance [30].

1. Level 1 is "Initial", where processes are not controlled and are unpredictable.
2. Level 2 is "Managed", where processes exist but are often reactive.
3. Level 3 is "Defined", where processes are standardized and typically documented.
4. Level 4 is "Quantitatively Managed", where processes are measured and controlled.
5. Level 5 is "Optimized", where processes have a focus on continuous improvement.

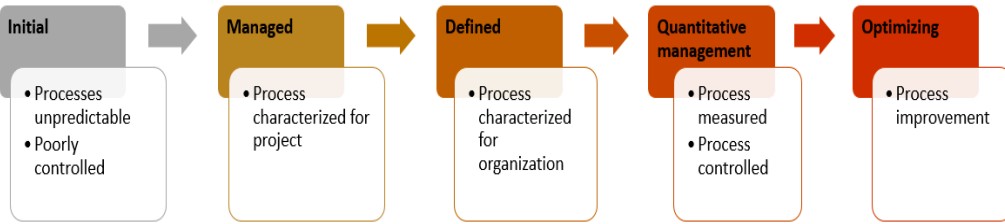

**Figure 2.** Capability maturity model integration (CMMI) [1,4,30].

The maturity model could be presented as a (1) "prescriptive model" responsible for the relationships between business performance and maturity improvement that can effectively influence the business value; (2) "descriptive model" that assesses the "as-is" state; or (3) "comparative model" that enables benchmarking within various areas or industries [11,34].

The maturity model consists of multi-dimensional levels of maturity regarding certain domains and can be used for organizational assessment and development [24]. It provides dimensions and characteristics that need to be achieved to reach a specific level of maturity. Through the maturity evaluation, the current state of the organization concerning the chosen criteria is defined. The criteria are evaluated to determine the maturity level of technology, organization, and people. The application of maturity models can be supported by predefined procedures such as an assessment questionnaire, checklist, or assessment tool [34]. Based on the as-is analysis recommendations for improvement to reach higher maturity levels, measures can be extracted and prioritized [34]. The probabilities of future profit for the organization can be determined by the current level of its Big Data in case the organization levels up its maturity model [35].

Regarding the Big Data maturity model, assuming how far data has come and how its velocity is changing, there is a clear need to measure Big Data's maturity [35]. The BDMM represents a roadmap that the organizations can adapt to guide their desirable efforts. It is also considered a classification tool for determining the status of an organization's Big Data and the required risk, cost, quality, and return on investment (ROI) values to achieve the desired levels [35]. In addition, the Big Data maturity model is considered a powerful tool that concentrates on organizational activity and delivers the best possible results for data collection, analysis, and visualization efforts [35]. The BDMM could also be used to assess organizational readiness, capability, technology, competence, success, and performance with relevance to Big Data across critical, predefined dimensions that would improve the organization's state of maturity [27,33].

The application of maturity models in Big Data is essential for governance and strategy implementations, because organizations need to assess their current maturity levels based on pre-identified criteria to effectively design a roadmap for achieving a higher level of maturity [31]. The degree of maturity defines a specific state of development within a range scaled and determined by an initial point (lowest development point) and an endpoint (highest development point). A particular level of maturity includes the specific characteristics of predefined objects and their requirements [34].

Big Data maturity assessment models are new concerns that require more study [33,36]. Maturity models help organizations identify the prerequisites to start the Big Data journey. In addition, maturity models can quickly position businesses across several criteria that help prioritize and plan for Big Data projects [36]. These activities should be applied in the organization by reporting and documenting the ongoing technology tools, resources, infrastructure, process, and business applications and determining how to improve their strategy [10,37]. The organization's current security should be assessed, such as recovery and backup systems, performance management, disaster recovery, and infrastructure monitoring processes. Organizations should complete the assessments, analyze the gaps, and improve their processes to ensure the compatibility of their technologies, people, and the organization itself and the maturity of readiness and implementation [10,28].

Big Data practitioners and statistics experts sometimes design maturity assessment models that can be very complex and hard to understand. Furthermore, the task may consume most of their organizational capabilities [4,33,38]. It has been found that the majority of the existing literature is mainly available as white papers or reports also registered on external developers' websites as internet materials [33,35,37]. These websites mainly reported on their success stories to promote their services in the business [28,39]. Technology providers or consulting partners have developed the existing models in the industry. They have developed several models with high biases to their organizational objectives and a low level of validation and evaluation, where their models suffer from

limited and inaccurate validity. On the subject of the maturity model, the majority of the available models were not following the standard levels of the capability maturity model (CMM), which are five levels provided by the Software Engineering Institute [29,30,33].

By conducting a systematic literature review (SLR) and applying a descriptive qualitative content analysis method, this paper attempts to address the limitations in the current literature and to be part of the research along with other researchers and practitioners in the available BDMM assessment dimensions and tools. This was achieved by providing answers to the three predefined research questions below:

RQ1: What are the existing maturity assessment models for Big Data?

RQ2: What are the assessment dimensions for Big Data maturity models?

RQ3: What are the assessment tools for Big Data maturity models?

Answering these predefined research questions could give managers critical insights to decide which tool fits within their organization and assess their Big Data maturity level. Moreover, the findings of these research questions could be used as a critical reference to propose a preliminary classification for the maturity assessment dimensions of Big Data that could be used to develop a new BDMM.

This SLR's structure starts with the Introduction in Section 1. The review methodology is presented in Section 2, highlighting the method used to conduct this paper. Next, the results are reported in Section 3 to recognize the existing Big Data maturity model, its assessment dimensions, and its assessment tools. Then, the limitations of existing Big Data maturity assessment models are described in Section 4. Finally, Section 5 concludes with some suggestions for future work.

## 2. Methods

This paper used the SLR method to determine the literature that focuses on Big Data maturity assessment models, their assessment dimensions, and tools. The SLR method is considered reliable; it is also suitable and accurate, making it suitable to evaluate the existing research related to a specific phenomenon of interest, research question, issue, or topic domain [39–41]. The systematic review strategy in this article was conducted based on the instructions for performing SLR in software engineering by [39,40] to answer the research questions. This systematic literature review contains the following main stages: (1) review planning, (2) review conducting (3) review reporting. The stages are presented in Figure 3.

### 2.1. Stage 1: Review Planning

The planning stage involves several steps, including pointing out the necessity for SLR, constructing research questions, and developing a review protocol that will construct the research question and the methods used to perform the review. Identifying the need for this SLR was highlighted in the previous section (Section 1). The limitations in the current literature were addressed in this SLR, and it also contributes to the body of knowledge of researchers and practitioners about the available Big Data maturity assessment models, assessment dimensions, and tools. The findings from this paper could help organizations identify the proper tool that fits with their organization and to know how to assess their maturity level for Big Data. This SLR identified three research questions (RQ), as presented in Table 1.

This SLR was performed through a predefined search strategy to identify the literature related to the research questions. The strategy used in this SLR aimed to pick out the main studies, including the resources and search keywords related to predefined research questions. The resources included conference proceedings, electronic search engines, gray literature, journals, and digital libraries. In this SLR, the planning stage evaluated the available research questions and findings. The conducting stage highlighted this systematic literature review's used sources and keywords. More details about the conducting stage will be found in the following sub-sections.

**Planning the review**

- Formulate the review's research question
- Develop the review's protocol

**Conducting the review**

- Search the relevant literature
- Perform selection of the primary studies
- Perform data extraction
- Assess studies' quality
- Conduct synthesis of evidence

**Reporting the review**

- Write up the SLR reports/papers

**Figure 3.** The systematic review phases.

**Table 1.** The SLR research questions and their contributions.

| ID | Research Questions | Contributions |
|----|--------------------|---------------|
| RQ1 | What are the existing maturity assessment models for Big Data? | To identify the existing maturity assessment models for Big Data. |
| RQ2 | What are the assessment dimensions for Big Data maturity models? | To identify the existing assessment dimensions for the existing Big Data maturity models. |
| RQ3 | What are the assessment tools for Big Data maturity models? | To identify the existing tools used to assess the maturity of Big Data. |

*2.2. Stage 2: Conducting the Review*

To conduct a systematic review, many stages should be applied, such as: identifying the search sources and the search strategy, in addition to the selection of main studies, extracting and monitoring the data and data synthesis, and studying the quality assessment [39,40]. As per [42], the lack of related studies represents the main validity threat. Some other examples of these threats are incorrect or automatic search, incomplete search terms, incorrect search method, inaccessible databases and papers, limited time duration, errors in the identification of main studies that occur during the search process, and finally, favoritism or bias.

The backward snowballing method was referenced by a list of related works that were first identified using the database search method. Both search methods were used during the search phase to avoid bias [42–44].

2.2.1. Sources

When constructing the review protocol, the appropriate databases and sources should be identified by determining all possible sources when conducting the stage of the review protocol [39]. This systematic literature review began with a database search method that identified the existing literature related to this SLR's study questions. As per [43] recommendation, we mainly used the database search method in this SLR as the first strategy. This SLR depended on electronic search engines and digital libraries as resources; conference proceedings, journals, and gray literature were also used [39]. After conducting a comprehensive search on several databases and search engines, seven resources were identified as the initial electronic databases for choosing the best literature related to the predefined research questions. The Tampere University of Technology has made a popularity-of-use list of best resources from which we chose eight databases for this

SLR [28], which were as follows: EBSCOhost, ScienceDirect, Scopus, Springer, Web of Science, Digital Library ACM, and IEEE. Publisher Elsevier owns both ScienceDirect and Scopus [28]. In order to eliminate any biased or redundant data, we dropped the database ScienceDirect. This SLR excluded the "Web of Science" from the main databases due to the limited availability of the related information. ResearchGate, Google, Google Scholar, and USM Library Repository were also chosen in order to have a manual search and included sources by the industry that could be found using the search engine Google.

Nine databases were finally selected as the list of search sources for this SLR. After applying the search strategy, it appeared that these databases had excellent repositories to show the available literature related to this SLR; as per the search, these databases were found to be the main sources of existing literature by other databases [44]. Using these key resources, we found published journal papers, conference proceedings, white/gray literature, IEEE bulletins, book chapters, white papers, symposiums, technology reports, workshops, and developers' websites. Other sources for this SLR search were used based on the backward snowballing method for the reference lists in the extracted papers.

### 2.2.2. Selection Criteria (Inclusion and Exclusion)

Study selection criteria are the criteria for determining whether to include or exclude a study from the systematic review [39].

The study used the exclusion and inclusion criteria for big data maturity assessment models in the reviewed selections. In this paper, the specified research questions were used to derive different combinations of search terms that were used to identify some keywords such as "Big Data", "Big Data Analytics", "Big Data Maturity Assessment", "Maturity Assessment", "Big Data Maturity Model", "Big Data Analytics Maturity Models", "Big Data Analytics Maturity Assessment Models", "Capability Assessment", "Maturity Model Framework", and "Big Data Maturity Assessment Models".

The most common technique used to identify a search strategy is to extract individual terms from the research question, which can be used to execute advanced search strategies by using Boolean "ORs" and "ANDs" [39,40]: for example, (Big Data OR Big Data Analysis, OR Big Data Maturity) AND (Maturity OR Maturity Assessment OR Maturity) AND (Big Data Maturity Assessment Model OR Big Data Maturity Model OR Big Data Maturity OR Big Data Analytics Maturity Models OR Big Data Capability Maturity Model). This research included Big Data maturity assessment models developed by academics and practitioners from 2007 to 2022 in English. Big Data is a novel concept that was not presented as an active research field before 2007; earlier publications about BD and BDMM could not be found [41]. This SLR added an additional time window to cover the published literature between 2007 and 2022.

Based on the selection criteria, identified papers must focus on the maturity models of Big Data or Big Data analytics. We excluded he research that did not meet the inclusion criteria. However, because of the lack of literature in this field, this SLR presented additional criteria to select related papers that focused on (A) Big Data maturity models by practitioners or researchers, (B) Big Data maturity models, Big Data analytics maturity models, or Big Data project maturity models, or (C) Big Data maturity models that assessed maturity levels or readiness levels. Additionally, several types of publications were reviewed in this SLR paper, such as content analyses, articles, meta-analyses, white papers, systematic literature reviews (SLR), surveys, case studies, and empirical case studies.

### 2.2.3. Quality Assessment

The quality assessment is an essential activity to assess the quality of the primary studies. Assessing the quality includes formulating predefined questions aimed at assessing how the chosen articles have addressed bias and external and internal validity [45]. The quality assessment questions (Q1–Q4) in this paper are shown in Table 2, and the answers were limited to three options: No = 0, Partially = 0.5, and Yes = 1.

**Table 2.** Study Quality Assessment Criteria.

| No. | Assessment Questions | Answer |
|-----|---------------------|--------|
| Q1 | Is there a clear description of the article's objective? | Yes/No/Partially |
| Q2 | Does the article adequately explain the assessment methods, dimensions, and tools? | Yes/No/Partially |
| Q3 | Is the article supported by primary data and material? | Yes/No/Partially |
| Q4 | Does the article clarify and detailed the model's constructs, dimensions, and structure? | Yes/No/Partially |

*2.3. Stage 3: Reporting the Review*

Reporting the review is one stage. It is important to communicate the results of a systematic review effectively [39]. The "discussion and findings" section discusses findings and results in the reporting stage.

**3. Discussion and Findings**

This SLR applied an expressive content analysis method that showed that the available papers did not cover every available BDMM developed by academics or practitioners. While industries with insufficient documentation have developed most of the existing BDMMs [28,33,35], limited publications investigated the available BDMMs.

The findings of this SLR aimed to answer the predefined research questions and address gaps in the existing literature. Five phases were used in the applied study selection process in this SLR: (1) A total of 359 related papers were chosen as possible sources from the digital search after filtering based on the initial keyword search results. After screening the results and extracting more related sources, (2) scanning by abstract, title and conclusion was applied to a total of 150 papers. (3) Before the quality assessment, a total of 75 articles in related studies were chosen to read their abstracts, introductory sections, and conclusions. (4) A total of 33 of the returned papers were extracted in the quality assessment after a complete review of the abstract and full text of these papers. (5) Irrelevant and duplicated articles were excluded using exclusion criteria by filtering the quality assessment stage results; then, a total of 15 articles were accepted and chosen as final sources for the data synthesis.

Based on the snowballing technique, the references extracted from the most related papers were checked manually to search for any other sources. Applying the full screening criteria after searching the keywords in the identified databases and removing the overlapping papers that were out of the research domain, this SLR selected 15 publications as the main sources for the existing BDMMs. The search process, results, and paper selection are shown in Figure 4.

There were a limited number of published papers in the literature per pear. It appeared that in 2013, the number of BDMM-related publications was 4 with same number of publications in 2014. However, it decreased in 2015 and 2016 to be 2 Publications in 2015, and 1 Publication in 2016. Also, there were one publication in each year of 2018–2020. However, there were no publications in the years of 2021, and 2022 (as shown in Figure 5). This SLR paper addresses the limitations in the existing literature and contributions to the BDMM field during the years 2007–2022. The studies and the ratio of publications per year are presented in Table 3.

After applying the content analysis method and performing quality assessments based on the given criteria, 15 articles were selected for this review. As presented in Table 4, the analysis rated 8 articles (53%) to be very good in terms of quality and 7 articles (47%) as good, eliminating the rest as poor quality articles.

Based on the predefined quality assessment questions (Q1–Q4), the outcomes (A1–A15) of the quality assessment applied to the 15 articles that were chosen for this SLR are shown in Table 5.

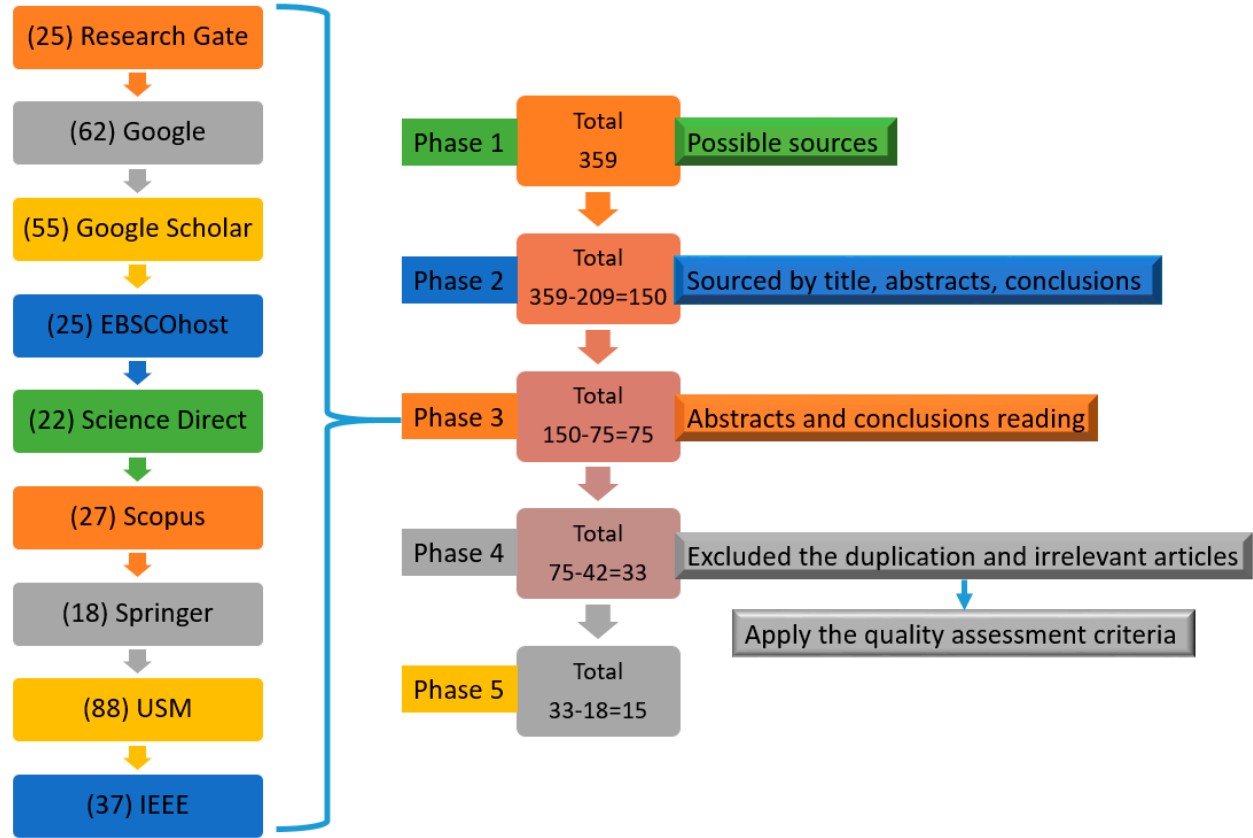

**Figure 4.** The Search process and results of paper selection.

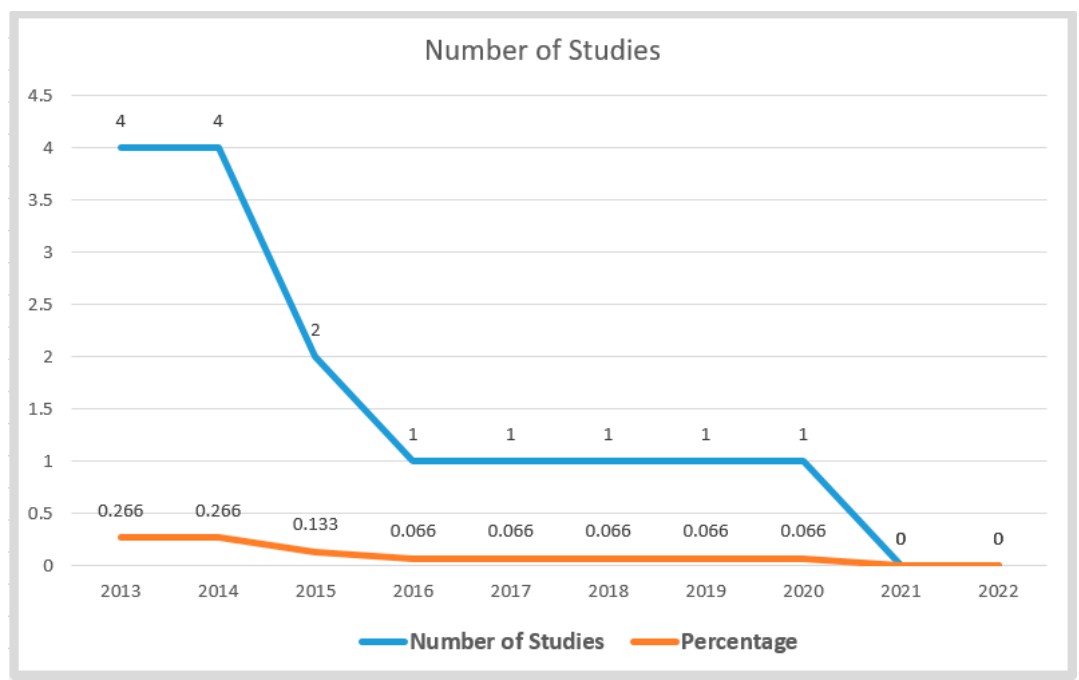

**Figure 5.** Distribution of BDMM studies by publication (2013–2022) year.

**Table 3.** Percentage of total studies published, by year.

| Year | Number of Studies | Percentage | References |
|---|---|---|---|
| 2013 | 4 | 0.266 | [38,45–48] |
| 2014 | 4 | 0.266 | [49–52] |
| 2015 | 2 | 0.133 | [42,43] |
| 2016 | 1 | 0.066 | [34,53] |
| 2017 | 1 | 0.066 | [44] |
| 2018 | 1 | 0.066 | |
| 2019 | 1 | 0.066 | |
| 2020 | 1 | 0.066 | [54] |
| 2021 | 0 | 0 | |
| 2022 | 0 | 0 | |

**Table 4.** Quality assessment score.

| Quality Scale | Very Poor (<1) | Poor (1–<2) | Good (2–<3) | Very Good (3–4) | Total |
|---|---|---|---|---|---|
| Number of papers | 0 | 0 | 7 | 8 | 15 |
| Percentage (%) | 0 | 0 | 47 | 53 | 100 |

**Table 5.** Quality assessment results.

| ID | Q1 | Q2 | Q3 | Q4 | Total |
|---|---|---|---|---|---|
| A1 | 1 | 1 | 0.5 | 1 | 3.5 |
| A2 | 1 | 0.5 | 0 | 1 | 2.5 |
| A3 | 1 | 1 | 0.5 | 1 | 3.5 |
| A4 | 1 | 1 | 0 | 1 | 3 |
| A5 | 1 | 0.5 | 0 | 1 | 2.5 |
| A6 | 1 | 1 | 0 | 0.5 | 2.5 |
| A7 | 1 | 1 | 0 | 0.5 | 2.5 |
| A8 | 1 | 1 | 0.5 | 0.5 | 3 |
| A9 | 1 | 1 | 0 | 0.5 | 2.5 |
| A10 | 1 | 0.5 | 1 | 1 | 3.5 |
| A11 | 1 | 1 | 0 | 1 | 3 |
| A12 | 1 | 1 | 0.5 | 0.5 | 3 |
| A13 | 1 | 1 | 0 | 0.5 | 2.5 |
| A14 | 1 | 1 | 0 | 0.5 | 2.5 |
| A15 | 1 | 1 | 1 | 1 | 4 |

After addressing gaps in this SLR, the following sub-sections reviewed and identified the existing BDMMs, assessment dimensions, and tools. This was done by providing answers to the following three research questions:

1. (RQ1): What are the existing maturity assessment models for Big Data?
2. (RQ2): What are the assessment dimensions for Big Data maturity models?
3. (RQ3): What are the assessment tools for Big Data maturity models?

The following sections present the existing Big Data maturity models. Next, the assessment dimensions of the existing models are identified, followed by a review of the assessment tools available in the existing models.

### 3.1. RQ1: What Are the Existing BDMMs?

Academics and practitioners have developed many maturity assessment models for different modifications of traditional domains to assess their maturity levels. With relevance to existing BDMMs, traditional maturity models have been modified [42]. However, there were not enough collaborative efforts for generalizing maturity model development in any area. Most of the accessible models did not sufficiently address the complexities of the issues of Big Data and have not been verified and evaluated in a real case study [29]. Most assessment tools and the Big Data maturity assessment models were designed for modeling experts, making them unsuitable for most organizations' front-line work. Hardly any manager believes that these models could be useful for their organization, technology, or capabilities [9,55]. These complex maturity models are not popular, have not attracted any interest or been adopted, and may lead to inaccurate and incorrect assessment results if the incorrect assessment model is applied [34].

Based on this SLR to analyze the existing BDMMs, only 15 publications were accepted and considered as final sources for the data synthesis relating to Big Data maturity models (10 Big Data maturity models by practitioners and five by academics). The origins, sources, levels, and dimensions of the existing BDMMs are shown in Table 6.

**Table 6.** Evaluation of the existing Big Data maturity models.

| ID | Assessment Model Name | Abbr. | Source | Origin | No. of Levels | Names of Levels | Maturity Dimensions |
|----|----|----|----|----|----|----|----|
| A1 | TDWI Big Data Maturity Model | TDWI BDMM | [45] | Practitioner-Educational-(TDWI) 2013 | 5 levels | Nascent<br>Pre-adoption<br>Early adoption<br>Corporate adoption<br>Mature/visionary | Data management, infrastructure, analytics, and organization governance |
| A2 | Big Data Business Maturity Model Index | BDBMMI | [46] | Practitioner (EMC) 2013 | 5 levels | Business monitoring<br>Business insights<br>Business Optimization data monetization<br>Business metamorphosis | Organization, business process, and organization's situation |
| A3 | IDC MaturityScape Big Data and Analytics | IDC MBDA | [47] | Practitioner (IDC) 2013 | 5 levels | Ad hoc<br>Opportunistic<br>Repeatable<br>Managed<br>Optimized | Intent, data technology, people, process |
| A4 | Maturity Model for Big Data Development | n/d | [48] | Practitioner (TNO) 2013 | 4 levels | Efficiency<br>Effectiveness<br>New solutions<br>Transformation | Data management, strategy, efficiency, effectiveness, new solutions, transformation, data and analytics, security and policy, and partnership |
| A5 | Enterprise Architecture Maturity Assessment tool | n/d | [38] | Practitioner (Infotech) 2013 | 4 levels | Undergo Big Data education<br>Assess Big Data readiness<br>Pinpoint a killer BD use case<br>Structure a Big Data proof-of-concept project | Technology, staffing, business focus, Big Data management and governance, data type and quality |
| A6 | Big Data Maturity Assessment | BDMA | [49] | Practitioner (Knowledgent) 2014 | 4 levels | Infancy<br>Technical adoption<br>Business adoption<br>Data and analytics as a service | Business need, technology platform, operating model, analytics, and information management |

**Table 6.** *Cont.*

| ID | Assessment Model Name | Abbr. | Source | Origin | No. of Levels | Names of Levels | Maturity Dimensions |
|---|---|---|---|---|---|---|---|
| A7 | Big Data Maturity Framework | BDMF | [50] | Practitioner (Booz & Company) 2014 | 4 levels | Performance management Functional area excellence Value proposition enhancement Business model transformation | Technical/infrastructure, data availability and governance, data-driven, decision-making culture, organization and resources, and sponsorship |
| A8 | Big Data Maturity Model | BDMM | [51] | Practitioner (Radcliffe Advisory Services) 2014 | 6 levels | In the dark Catching up First pilot(s) Tactical value Strategic leverage Optimize and extend | Vision, strategy, value and metrics, governance, trust and privacy, people and organization, data sources, data management, and analytics and visualization |
| A9 | A Maturity Model for Big Data and Analytics IBM | MMBDA | [52] | Practitioner (IBM)-2014 | 4 levels | Ad hoc Foundational Competitive differentiating Breakaway | A business strategy, information, analytics, culture and operational execution, architecture and governance |
| A10 | Zakat Big Data Maturity Model | ZBDMM | [43] | Academia-2015 | 5 levels | Ignorance Coping Understanding Managing Innovating | Organization, leadership, data governance and integration, and analytics |
| A11 | The Big Data Temporal Maturity Model | BDTMM | [42] | Academia-2015 | 5 Levels | Atemporal | Data/knowledge |
| | | | | | | Pre-temporal | IT solutions |
| | | | | | | Partly temporal | functionalities |
| | | | | | | Predominantly temporal | |
| | | | | | | Temporal | |
| A12 | Hortonworks Big Data Maturity Model | n/d (Hortonworks model) | [53] | Practitioner (Hortonworks) Internal-2016 | 4 levels | Aware Exploring Optimizing Transforming | Sponsorship, data and analytics, technology and infrastructure, organization and skills; and process management |
| A13 | Big Data Maturity Model by Comuzzi | BDMM | [34] | Academia-2016 | 6 levels | Non-Existent (Awareness) Initial Repeatable Defined Managed Optimized | Strategic alignment, data, organization, governance, information technology |
| A14 | A Value-Based Big Data Maturity Model | n/d | [44] | Academia-2017 | 5 levels | Initial (Pre-contemplation) Defined (contemplation) Managed (preparation) Optimized (commitment) Strategic (future) | Organization, governance, data management, strategy, value and metrics, and trust and privacy |
| A15 | A maturity model for big data analytics in airline network planning | n/d | [54] | Academia-2020 | 6 levels | n/d | Strategic alignment, organization, data, information technology |

Source: SLR and compilation by author.

The Data Warehousing Institute (TDWI) proposed and developed the first model of Big Data maturity. The aim was to assess the maturity of a Big Data and Big Data analytics program over several dimensions, such as data management, analytics, infrastructure, organization, and governance, which was considered a solution to benefit from analytics of Big Data [45]. This model aimed to describe the organization's ability to benefit from Big Data value, which could be achieved by pursuing the activities and stages for adoption of Big Data initiatives and comparing themselves against others based on such efforts [42,45]. The TDWI Maturity Model is considered a guide and roadmap that provides a self-assessment tool based on the model [42]. Furthermore, the TDWI Big Data Maturity Model assessment tool objectively measures the maturity of an organization's Big Data and Big Data analytics program across the model's dimensions [45]. The TDWI Big Data Maturity Model is like the Business Intelligent (BI) model, which consists of five successive stages: nascent, pre-adoption, early adoption, corporate adoption, and mature/visionary, as shown in Figure 6. Organizations should move through these stages to gain more value from their investments [42,45].

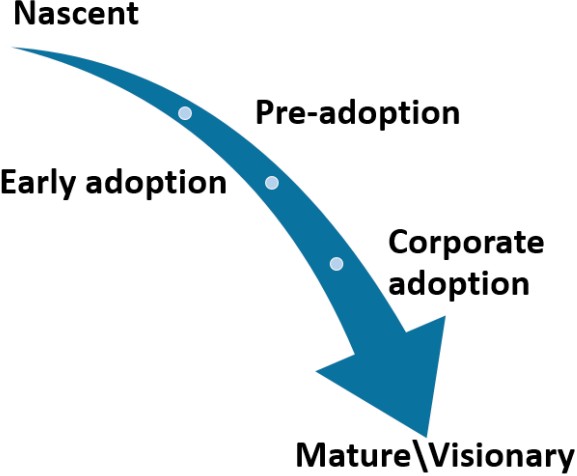

**Figure 6.** The five stages of the TDWI Maturity Model.

The main dimensions that characterize the TDWI (Infrastructure, Data Management, Analytics, Governance, and Organization) are illustrated in Figure 7:

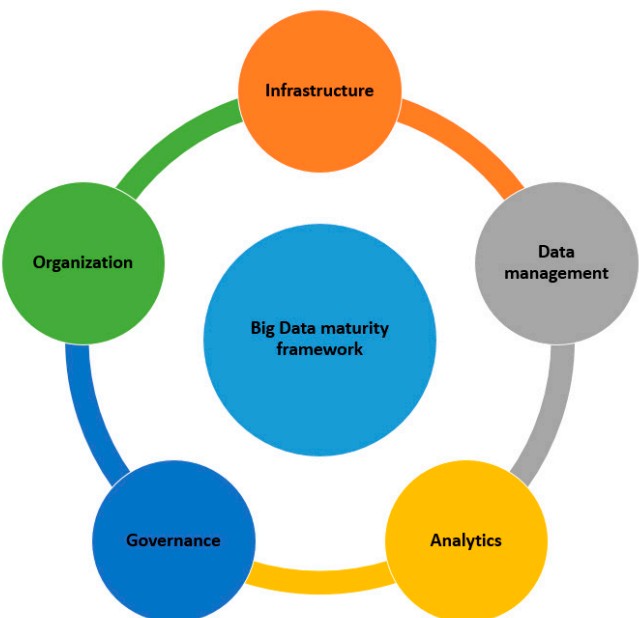

**Figure 7.** The dimensions of the TDWI Maturity Model.

Other factors that characterize the TDWI are illustrated in Figure 8.

**Figure 8.** The other dimensions of the TDWI Maturity Model.

The TDWI Model provides 50 benchmarking questions from the TDWI website for organizations to assess their maturity levels [30]. The studies by [36,40] highlighted that only the TDWI was documented and originated by the practitioner based on educational background. Moreover, it is concerned with the entire organization and its processes, not only with the IT infrastructure.

In 2013, a study by [46] presented the Big Data Business Model Maturity Index (BDBMMI). This index is used to assess business model maturity on the subject of Big Data, which helps organizations measure their effectiveness at leveraging data and analytics to power their business models [42,56]. Five stages were proposed in the BDBMMI, which are as follows: Business Monitoring, Business Insights, Business Optimization, Data Monetization, and Business Metamorphosis. The first three stages of this maturity model focus on the organization's internal and optimizing internal business processes. The last two are focused on the organization's environment. The Big Data Business Model Maturity Index was developed based on four critical dimensions: strategy, analytics, business processes, and IT infrastructure [42,56]. Figure 9 shows the Big Data Business Model Maturity Index Levels.

Based on this SLR, a limited documented study clearly revealed the dimensions, subdimensions, and validation method used in the BDBMMI. The available details regarding this model are based only on one existing survey paper [42], website contents, and a limited-access chapter in a book by [42,56].

IDC MaturityScapes was created by International Data Corporation (IDC) to evaluate the competency and maturity of an organization's Big Data analytics (BDA), in addition to explaining Big Data adoption stages that start with the simple stage: unstructured, ad hoc and ending with the systematized and advanced level. The IDC model was provided by [47]. This model focuses on the key dimensions used to help management use Big Data in business: technology, people, processes, culture, and data [57]. According to [58], IDC's Big Data Analytics Maturity Model can help organizations to prioritize their resources in terms of their critical dimensions.

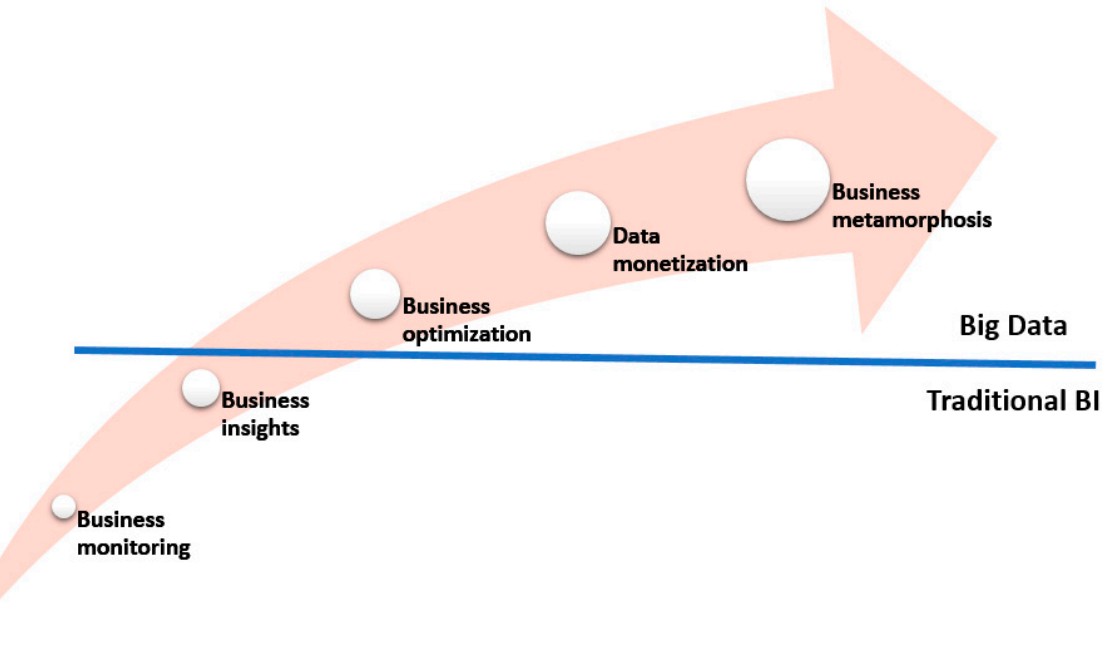

**Figure 9.** Big Data Business Model Maturity Index.

IDC MaturityScapes use the same pattern as most maturity models by following five stages (Ad hoc, Opportunistic, Repeatable, Managed, Optimized) that represent a progression from disorganization (ad hoc) to a highly systematized environment (optimized) [58], as shown in Figure 10.

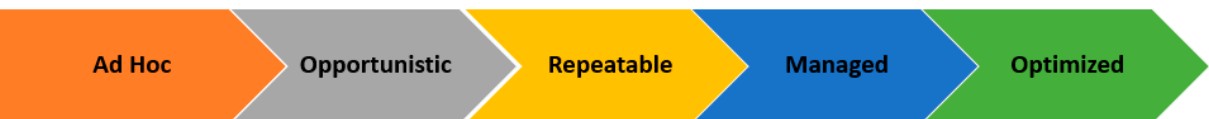

**Figure 10.** IDC MaturityScapes Stages.

The IDC's Big Data Maturity Model was derived from an IDC research paper published by [47], and it was transferred to a software tool hosted by Computer Sciences Corporation (CSC) [36].

In 2014, a BDMM was proposed by Advisory Services [51] for estimating the ongoing ("As Is") Big Data maturity and creating a vision that leads to achieving the ("To-Be") Big Data state covering eight capabilities. Those capabilities are people, strategy, analytics, data management, visualization, IT security policy, metrics, and vision [42,51]. The BDMM model is constructed from six levels (level 0—In the Dark, level 1—Catching Up, level 2—First Pilot(s), level 3—Tactical Value, level 4—Strategic Leverage, and level 5—Extend and Optimize), where the additional level (0—In the Dark) represents the initial level when organizations are unaware of opportunities, requirements, and challenges regarding Big Data [51]. The Radcliffe model is considered a general model; however, it does not propose any confirmation of the model, dimensions, or self-evaluation tools. It provides signs of Big Data initiatives that help organizations level their sequential maturity [42].

In addition to the previous models, ref. [34] proposed a Big Data Maturity Model to help organizations leverage Big Data and its added value. In [34], the model's name is "Big Data Maturity Model". BDMM consists of six stages (1—Non-Existent (Awareness), 2—Initial, 3—Repeatable, 4—Defined, 5—Managed, 6—Optimized). This model focused on key dimensions of strategic alignment, data, organization, governance, and information technology. A qualitative approach was used to develop BDMM, and the approach was based on semi-structured interviews with domain experts and literature analysis to

assess the implications of using the Big Data technology for business. The practitioners qualitatively evaluated the usefulness and completeness of this model, while Big Data maturity assessments evaluated the applicability of the model [34]. The privacy and security domains are some of the limitations in BDMM, in addition to the Big Data characteristics relevant to Big Data maturity and readiness, which require more investigation [34].

Another study by [42] presented temporal BDMM with limited application. The model by [42] was evaluated qualitatively to assess the readiness for Big Data. This model consists of three dimensions: Data/knowledge, IT solutions, and Functionalities. Additionally, it followed five stages (1—Atemporal, 2—Pre-Temporal, 3—Partly Temporal, 4—Predominantly Temporal, 5—Temporal).

Another proposed maturity model that focused mainly on managing the data quality for Big Data was presented in [44], considering the importance of data quality in business. According to [44], competitive business advantages are transformed in the era of Big Data, to compete for data quality and enabling the most powerful analysis tools to transform the data into information and knowledge for more competitive advantage. This model followed five stages (1—Initial (Pre-Contemplation), 2—Defined (Contemplation), 3—Managed (Preparation), 4—Optimized (Commitment), and 5—Strategic (Future)). Moreover, it consists of six dimensions (organization, governance, data management, strategy, value and metrics, and trust and privacy).

Another study, by [54], focused on a maturity model for big data analytics in airline network planning, proposed a big data maturity model for the airline industry. The model by [54] had six maturity levels and four main domains (Strategic Alignment, Organization, Data, and Information Technology). The development of this maturity model was grounded in the literature, expert interviews, and case study research involving nine airlines. Four airline business models were represented: full-service carriers, low-cost airlines, scheduled charter airlines, and cargo airlines. The maturity model has been well received, with seven change requests in the model development phase.

The existing maturity models have been designed to analyze the maturity of Big Data or readiness for Big Data based on a comprehensive set of dimensions/criteria [29]. These dimensions/criteria will be evaluated in the next section. All of the above model types, abbreviations, purposes, focus areas of the assessment models, capability components (dimensions), and tools are summarized in Table 7, which clarifies the comparison between the available BDMMs.

**Table 7.** The current Big Data maturity models.

| ID | Assessment Model Name | Abbr. | Primary Source | No. of Levels | Assessment Instrument/ Tool | Assessment Approach \Scale | Purpose of Use | Purpose of the Model Used | Focus Domain | Focus Area | Reliability and Validity of Assessment | Assessment Components |
|---|---|---|---|---|---|---|---|---|---|---|---|---|
| A1 | TDWI Big Data Maturity Model | TDWI BDMM | [45] | 5 levels | Software Tool | Qualitative and quantitative assessments | Comparative | To describe the maturity stages of an organization's capabilities and readiness for Big Data development | Big Data readiness | Big Data maturity and organization readiness | Validated (using benchmark survey) | Data management, infrastructure, analytics, organization and governance. |
| A2 | Big Data Business Maturity Model Index | BDBMMI | [46] | 5 levels | Text Document | Qualitative assessment | n/d | To measure the maturity of business models in the context of using Big Data and analytics. | Organization readiness (Big Data business model) | Organization readiness (business model) | Verified | Organization, business processes, and organization's situation. |
| A3 | IDC MaturityScape Big Data and Analytics | IDC (2013) MBDA | [47] | 5 levels | Text Document | Quantitative assessment | Comparative | To assess organization's competencies to leverage and manage BDA solutions. | Big Data and Analytics | Big Data analytics maturity and organization readiness | Verified | Intent, data, technology, people, processes |
| A4 | Maturity Model for Big Data Developments | n/d | [48] | 4 levels | Text Document | N/A | Prescriptive | Prescriptive to assess their own Big Data maturity and innovation potential | Big Data | Organization's capability or readiness | N/A | Data management, strategy, efficiency, effectiveness, new solutions, transformation, data and analytics, security and policy, and partnership |
| A5 | Enterprise Architecture Maturity Assessment tool | n/d | [38] | N/A | Software Tool | N/A | Prescriptive | To assess enterprise architecture maturity | Big Data | Limited to the operational and value perspective | N/A | Technology, staffing, business focus, Big Data management and governance, data type and quality |
| A6 | Big Data Maturity Assessment | BDMA | [49] | 4 levels | Software Tool | Quantitative assessment | Descriptive | To provide an assessment tool for an organization's Big Data maturity across five key dimensions. | Organization readiness for Big Data | Organization readiness | Verified | Business need, technology platform, operating model, analytics, and information management. |

**Table 7.** *Cont.*

| ID | Assessment Model Name | Abbr. | Primary Source | No. of Levels | Assessment Instrument/ Tool | Assessment Approach \Scale | Purpose of Use | Purpose of the Model Used | Focus Domain | Focus Area | Reliability and Validity of Assessment | Assessment Components |
|---|---|---|---|---|---|---|---|---|---|---|---|---|
| A7 | Big Data Maturity Framework | BDMF | [50] | 4 levels | Text Document | Qualitative assessment | Prescriptive | To categorize the numerous ways in which data can be an advantage, from selective adoption to large-scale implementation. | Organization readiness for Big Data | Organization readiness | Verified | Technical/infrastructure, data availability and governance, data-driven, decision-making culture, organization and resources, and sponsorship. |
| A8 | Big Data Maturity Model | BDMM | [51] | 6 levels | Text Document | Qualitative assessment | Prescriptive | To socialize the concepts and critical success factors around Big Data maturity, assess the level of existing Big Data maturity, and then build a Big Data vision and roadmap. | Big Data maturity | Effectiveness of Big Data adoption and implementa-tion | Verified | Vision, strategy, value and metrics, governance, trust and privacy, people and organization, data sources, data management, and analytics and Visualization |
| A9 | A Maturity Model for Big Data and Analytics IBM | MMBDA | [52] | 4 levels | Text Document | Quantitative assessment | Descriptive | To provide a guide on identifying business value using Big Data and analytics. | Big Data and analytics (business model) | Business model | Verified | A business strategy, information, analytics, culture and operational execution, architecture and governance. |
| A10 | Zakat Big Data Maturity Model | ZBDMM | [43] | 5 levels | Text Document | Qualitative assessment | n/d | To gauge the readiness of zakat institutions to embark on a Big Data evolution. | Big Data | Organization readiness for a non-profit organization | Verified | Organization, leadership, data governance and integration, and analytics. |
| A11 | The Big Data Temporal Maturity Model | BDTMM | [42] | 5 Stages | Assessment Tool and Question-naire | Qualitative assessment | n/d | To assess the readiness for Big Data | Big Data | Organization readiness for Big Data | | Data/knowledge, IT solutions, Functionalities |

**Table 7.** *Cont.*

| ID | Assessment Model Name | Abbr. | Primary Source | No. of Levels | Assessment Instrument/ Tool | Assessment Approach \Scale | Purpose of Use | Purpose of the Model Used | Focus Domain | Focus Area | Reliability and Validity of Assessment | Assessment Components |
|---|---|---|---|---|---|---|---|---|---|---|---|---|
| A12 | Hortonworks Big Data Maturity Model | n/d | [53] | 4 levels | Scorecard Survey | Qualitative assessment | n/d | To provide a guide and roadmap for assessing the current state of Big Data maturity | Big Data | Business transforma-tion | Verified (based upon previous consulting experiences) | Sponsorship, data and analytics, technology and infrastructure, organization and skills, and process management |
| A13 | Big Data Maturity Model | BDMM | [34] | 6 levels | Text Document | Qualitative assessment | n/d | To assess Big Data maturity | Big Data | Business implication | Verified | Strategic alignment, data, organization, governance, information technology |
| A14 | A Maturity Model for Big Data and Analytics IBM | BDMM | [44] | 5 levels | | NA | n/d | Proposed a value-based maturity model | Big Data value | Focuses only on the data quality management of Big Data | NA | organization, governance, data management, strategy, value and metrics, trust and privacy |
| A15 | A maturity model for big data analytics in airline network planning | MM | [54] | 6 levels | Online Survey | Qualitative research approach | Comparative | Proposed a maturity model for big data analytics in airline network planning | Big Data analytics in airline network planning | maturity model for Big Data readiness for airline network planning | Verified | Strategic alignment, organization, data, information technology |

Source: SLR and compilation by author and from Refs. [24,25,27,29,35,48,50].

### 3.2. RQ2: What Are the Assessment Dimensions for Big Data Maturity Models?

It is important to consider the details of the characteristics that existing BDMMs have shown relative to each other [27,34,35]. The main criteria differentiating the BDMMs from the rest are their capability elements (dimensions/criteria). These describe the components/elements of the Big Data ecosystem included in the assessment and can scope and summarize the capability elements, including organization, technology, data, processes, system architectures, and people [35].

Although the Big Data maturity models are very similar, the assessment methods and the dimensions used in the maturity assessment are different; those dimensions and methods include self-assessment, internal assessment, and external assessment. Third parties and vendors have conducted the assessments, as well as certified practitioners with commercial intent [36].

In the maturity model domain, a comprehensive literature review was applied to identify assessment dimensions [29]. The results of this SLR show that the existing articles might not have all covered the available dimensions and capabilities of the Big Data maturity assessment models.

The maturity assessment dimensions could be identified by investigating domain-specific critical success factors (CSFs) and demanding the data collection methods, for example, interviews, official group technique, Delphi method, focus groups, and case studies [36]. Specifically, critical success factors (CSFs) and challenges to provide valued insights into domain elements (dimensions) were indicated by [59]. The available BDMMs did not identify the sources of their assessment dimensions. Additionally, they did not identify their data collection or analysis methods.

The existing literature (from 2010 until 2022) contains no clear documentation or referential documents for Big Data maturity assessment, as most of the available models were from vendors or IT players still on their websites or blogs, and there are no academic papers or models developed by academics for the purposes of validity and reliability. The studies by [36] and [34] supported the findings from our SLR in this section, namely that the available maturity models do not cover the full critical domains and dimensions that Big Data maturity models should consider. That calls for developing a new BDMM that covers the critical dimensions relevant to Big Data maturity assessment. Based on a descriptive and qualitative content analysis, the critical dimensions that differentiate the BDMMs and their frequencies in the literature are presented in Table 8.

**Table 8.** The dimensions of Big Data maturity and the respective literature.

| BD Maturity Dimensions | Existing BDMMs | | | | | | | | | | | | | | |
|---|---|---|---|---|---|---|---|---|---|---|---|---|---|---|---|
| | A1 | A2 | A3 | A4 | A5 | A6 | A7 | A8 | A9 | A10 | A11 | A12 | A13 | A14 | A15 |
| | [45] | [46] | [47] | [48] | [38] | [49] | [50] | [51] | [52] | [43] | [42] | [53] | [34] | [44] | [54] |
| Data Management | ✓ | | | ✓ | | | | ✓ | | | | | | ✓ | |
| Big Data Management | | | | | ✓ | | | | | | | | | | |
| Data Type and Quality | | | | | ✓ | | | | | | | | | | |
| Information Management | | | | | | ✓ | | | | | | | | | |
| Data-Driven | | | | | | | ✓ | | | | | | | | |
| Trust and Privacy | | | | | | | | | | | | | | ✓ | |
| New IT Solutions | | | | ✓ | | | | | | | ✓ | | | | |
| Transformation | | | | ✓ | | | | | | | | | | | |
| Infrastructure | ✓ | | | | | | | | | | | | | | |
| Technology | | | ✓ | | ✓ | | | | | | | | | | |

**Table 8.** *Cont.*

| BD Maturity Dimensions | Existing BDMMs | | | | | | | | | | | | | | |
|---|---|---|---|---|---|---|---|---|---|---|---|---|---|---|---|
| | A1 | A2 | A3 | A4 | A5 | A6 | A7 | A8 | A9 | A10 | A11 | A12 | A13 | A14 | A15 |
| | [45] | [46] | [47] | [48] | [38] | [49] | [50] | [51] | [52] | [43] | [42] | [53] | [34] | [44] | [54] |
| Technology Platform | | | | | | √ | | | | | | | | | |
| Technology and Infrastructure | | | | | | | √ | | | | | √ | | | |
| Information Technology | | | | | | | | | | | | | √ | | √ |
| Architecture | | | | | | | | | √ | | | | | | |
| Process | | | √ | | | | | | | | | | | | |
| Business Process | | √ | | | | | | | | | | | | | |
| Data Sources | | | | | | | | √ | | | | | | | |
| Process Management | | | | | | | | | | | | √ | | | |
| Operating Model | | | | | | √ | | | | | | | | | |
| People | | | √ | | | | | | | | | | | | |
| Staffing | | | | | √ | | | | | | | | | | |
| Analytics | √ | | | | | | √ | | √ | √ | | | | | |
| Analytics and Visualization | | | | | | | | √ | | | | | | | |
| Data and Analytics | | | | √ | | | | | | | | √ | | | |
| Data | | | √ | | | | | | | | √ | | √ | | √ |
| Information | | | | | | | | | √ | | | | | | |
| Organization | √ | √ | | | | | | | | √ | | | √ | √ | √ |
| Organization's Situation | | √ | | | | | | | | | | | | | |
| Vision | | | √ | | | | | √ | | | | | | | |
| Strategy | | | | √ | | | | √ | | | | | | √ | |
| Strategic Alignment | | | | | | | | | | | | | √ | | |
| Efficiency | | | | √ | | | | | | | | | | | |
| Effectiveness | | | | √ | | | | | | | | | | | |
| Business Focus | | | | | √ | | | | | | | | | | |
| Business Need | | | | | | √ | | | | | | | | | |
| Business Strategy | | | | | | | | | √ | | | | | | |
| Partnership | | | | √ | | | | | | | | | | | |
| Decision-Making Culture | | | | | | | √ | | | | | | | | |
| Organization and Resources | | | | | | | √ | | | | | | | | |
| Sponsorship | | | | | | | √ | | | | | √ | | | |
| Value and Metrics, | | | | | | | | √ | | | | | | √ | |
| Culture and Operational Execution | | | | | | | | | √ | | | | | | |
| Leadership | | | | | | | | | | | √ | | | | |
| Organization and Skills | | | | | | | | | | | | √ | | | |
| People and Organization | | | | | | | | √ | | | | | | | |
| Governance | √ | | | | √ | | | √ | √ | | | | √ | √ | |
| Data Governance and Integration | | | | | | | | | | √ | | | | | |

**Table 8.** *Cont.*

| BD Maturity Dimensions | Existing BDMMs | | | | | | | | | | | | | | |
|---|---|---|---|---|---|---|---|---|---|---|---|---|---|---|---|
| | A1 | A2 | A3 | A4 | A5 | A6 | A7 | A8 | A9 | A10 | A11 | A12 | A13 | A14 | A15 |
| | [45] | [46] | [47] | [48] | [38] | [49] | [50] | [51] | [52] | [43] | [42] | [53] | [34] | [44] | [54] |
| Security and Policy | | | | √ | | | | | | | | | | | |
| Data Availability and Governance | | | | | | | √ | | | | | | | | |
| Trust and Privacy | | | | | | | | √ | | | | | | | |
| Intent | | | √ | | | | | | | | | | | | |
| Functionalities | | | | | | | | | | | | √ | | | |
| Strategic Alignment | | | | | | | | | | | | | | | √ |

*3.3. RQ3: What Are the Assessment Tools for Big Data Maturity Models?*

The maturity model application represents the physical conversion as a proof-of-concept of prior artifacts. Pre-identified procedures such as a questionnaire can support the application of maturity models. Based on the results of analyzing the current state (as-is state), guidelines and recommendations will be derived and prioritized to improve the results and reach a higher level of maturity [60].

A traditional or software-based assessment questionnaire can be developed from assessment instruments; every identified dimension can have formulated control assessment questions [36,60,61]. It is recommended to use electronic quantitative data collection methods because they increase the availability, generalizability, and applicability of the maturity model [29]. The number of questions in the assessment instrument must be balanced to ensure all domains are enveloped and the responses remain reliable [29,36]. Three approaches can be featured: self-assessment, third-party assisted, or certified professional-assisted [29,60]. Self-assessment tools are often not accessible due to the commercial intent of vendors [36,60]. Furthermore, how the assessment instrument will be used should be identified.

The studies by [45,47,49] offered their Big Data maturity assessment instruments as software tools. In all three of these models, the software assessment tool automatically calculates a maturity score based on the answers to a certain number of questions. The study by [38] offered their assessment instrument as a traditional questionnaire, and the calculation is performed with the help of spreadsheet functionality. This type of traditional questionnaire is not as effective as the software assessment. The rest of the existing models presented their maturity models as text documents, not providing an assessment directly to the end-user. This means that the organizations have to assess themselves and figure out the best way to utilize these models' descriptive and prescriptive content to assess their Big Data maturity and capabilities [36]. Big Data maturity could be assessed by developing a questionnaire tool to assess Big Data maturity across various dimensions [29].

When examining the visualization of the available Big Data maturity models, the model in [47] was the only one that interactively built its visualization. A visual chart is built for every business dimension as well as an overall score and alignment score. These charts can then be modified by interacting with specific parameters. Refs. [45,48–51] all illustrated their maturity models as traditional figures, helping the end-user to understand and adapt the basic concepts of the models quickly. No visualization was identified for the models in [38,52], both presenting their models as only textual [36]. As different maturity assessment models already exist, it is necessary to compare these existing models to find and justify the most suitable framework for Big Data maturity assessment. Then, a questionnaire tool would be developed as a quantitative method for assessing the Big Data maturity [29]. Among the available BDMMs, the types of assessment instruments that

are used to visualize and support the respective maturity assessment models are shown in Table 7.

## 4. The Limitations of the Available BDMMs

In 2013, McKinsey Global Institute (MGI) noted that sometimes, big data models are designed in a way that makes them very complex and capability-consuming for organizations [9,34–55]. Most of the tools were designed for specialists, not front-line organization workers. Most managers think that the existing models cannot work for the organization and cannot fit its existing capabilities [9,55]. Due to the importance of Big Data maturity, this requires additional investigation [34].

According to [36], there are obvious differences in the comprehensive performance of the available BDMMs. After reviewing the previous studies related to maturity models, the results indicated that the models of [45] and [47] provide end-users with all descriptive, prescriptive, and comparative functionalities, while the models of [38,48,50,62] serve a descriptive or prescriptive purpose of use. The models of [49,52] were the only ones acting as descriptive models, not providing any recommendations or improvement activities. Upon investigation, it was also noted that none of the models were structured as CMMs, but just as maturity grids or Likert-scale questionnaires. CMMs provide the right amount of complexity, also defining specific goals for key process areas and considering common implementation and infrastructural activities [36], which are not seen in the constructs of most of the existing models.

Comprehensive research by [28,35] evaluated and benchmarked eight available BDMMs that included models presented by [38,45,47–52]. Another study by [34] compared BDMMs that were available in the literature until the date of their research in 2016 as [38,45,49–52]. According to [24,46,47], the maturity models are missing many details such as documentation and dimensions, and also, no details are available about the process development or model validation and evaluation; hence, the models' internal validity is considered limited. Furthermore, the available models that were evaluated by previous studies were only being promoted by technology vendors, consulting companies, or professional education providers, and they did not guarantee an unbiased and equitable academic view of the opportunities provided by Big Data promises.

According to studies by [34–36,42], the top models presented by [45] and [47] contain validated and maintained design methods in their available maturity model documentation. Two models (TDWI and IDC) were determined to be the most effective by the study in [36]; those models cover the critical dimensions of organization, infrastructure, governance, analytics, and data management as essential elements for maturity assessment. A benchmarking study by [36] showed how the two models obtained high scoring in all criteria requirements, which enabled them to surpass the rest. TDWI and IDC MaturityScapes Stages models are rated as top-performing models. The two models are also considered the most useful for Big Data maturity assessments of quality and business value creation [36].

Based on an analysis of the existing models, the models by TDWI [45] and IDC [47] were considered effective models by IT practitioners with clear components that followed CMM standards. From the academic side, a study by [43] presented an effective model that follows the standards of five CMM levels, with a focus area on Zakat in the case study of a non-profit organization. The maturity model by [44] focuses on the assessment of Big Data quality management and is considered a good-quality reference for Big Data maturity models.

BDMMs focus on maximizing added value if the organizations level up their maturity models [34,44]. Other investigations have revealed how the majority of the current models are limited to Big Data domains and also are not adopting the five standard CMM maturity levels presented by SEI [34–36].

The study by [34] highlighted that only [47] and [51] recognized maturity levels somehow aligned to the capability maturity models' standard levels. The model proposed by [49] determines the maturity model according to the penetration of Big Data technology in the organization, while other models define maturity using ad hoc defined names. The

findings by [34] showed that only seven Big Data maturity models—TDWI, BDBMMI, IDC, BDMM, MMBDA, ZBDMM, and BDMM—present five levels of assessment that begin with the infancy stage until the organization is ready and mature for Big Data adoption, whereas the maturity models for Big Data development—BDMA, BDMF, and Hortonworks BDMM—have four levels of maturity assessment. Regarding the scale types, usually qualitative or quantitative assessments are used in the maturity models. The TDWI model applies both approaches (qualitative and quantitative assessment approaches). However, six of the Big Data maturity models (BDBMMI, BDMF, BDMM, ZBDMM, Hortonworks Big Data Maturity Model, and BDMM) use only a qualitative assessment approach, while the IDC BDMA and MMBDA use a quantitative approach.

The assessment instruments used in [38,45,49] are software tools, while the rest of the models used only text documents as instruments for assessment. The source documents showed that the previously related models (TDWI, BDBMMI, IDC Maturity Model for Big Data Development by [48], BDMA, BDMF, and ZBDMM) were developed or constructed to assess the maturity of Big Data itself after implementation or to assess the maturity of Big Data development. Another BDMM, by [38], is limited to assessing the maturity of Big Data governance. In addition, the model in [34] was developed only to assess the maturity of the business implications of Big Data. Ref. [35] indicated that six of these models (as presented in [43,45–47,49,50]) provide for assessments of Big Data development maturity and also can be used to assess the current state and desired state of readiness. However, the assessments ignored the ability to determine the required personnel competencies and skills relevant to Big Data.

The study by [35] found that the available models are more suitable for assessing the maturity of an organization's readiness. BDMMs have widened their purpose to include the assessment of Big Data implementation. The studies by [34–36,42] discovered that the studied assessment models have some critical limitations, such as poor documentation to guide organizations, and most of the assessment models are limited in scope to the maturity of Big Data itself.

Consequently, we conclude that the available BDMMs need more investigation. Based on the results of this SLR, we recommend TDWI and IDC's Big Data maturity models as candidates for use, as they fit the described feature criteria. The TDWI model scored a 3.5 in quality assessment; the IDC MaturityScapes Stages model also scored 3.5 (out of 5.0). IDC and TDWI's Big Data models are two that use different domains. The IDC model comprises intent, data, technology, people, and process. The Halper and Krishnan TDWI model includes distinct attributes of organization, infrastructure, data management, analytics, and governance.

Another investigation regarding model document types showed that BDMMs by [38,45–53] used white papers and practitioners' websites to publish their maturity models. The internet materials and papers are handbooks that guide users when identifying the Big Data maturity levels of their organizations. Both [38] and [52] did not provide any supporting materials or present all information within the frames of their maturity models [36].

Content analysis methods for these web materials were applied to identify the dimensions and levels of their models. The models by [52] and [38] did not provide any primary data or materials, nor did they present their constructs or information about measurements for their maturity assessment models.

Recently, some researchers have used BD models to tackle COVID-19 problems, such as in [63], which studied how nations are using machine learning and Big Data analytics to fight COVID-19. In [64], Big Data and artificial intelligence applications were studied in the battle against the COVID-19 pandemic. In [65], building and managing smart cities was studied using digital twins and BIM Big Data according to the COVID-19 concept. Ref. [66] presents a Big Data Bayesian network graph model for real-time Twitter stream identification with COVID-19. To maintain SME supply chain operations in the post-COVID-19 scenario, Big Data-driven creativity is suggested [67], together with the

moderating function of SME technology. In [68], the authors classified and studied people's mental states in order to spread awareness of mental health, particularly during the COVID-19 pandemic. Changes in primary care visits arising from the COVID-19 pandemic were studied in [69] with an international comparative study by the International Consortium of Primary Care Big Data Researchers. This study [70] used cognitive networks with the Anticipation, Logistics, Conspiracy, and Loss of Trust models to extract information on COVID-19 vaccines from popular English and Italian tweets. In another study [71], an interdisciplinary framework for a research paper was presented. This study looked at the theoretical underpinnings and research frameworks explaining the stability and outcomes of Big Data analytics.

Moreover, most of the available assessment models lack the assessment instruments, tools, and visualizations for assessment results, such as software tools to support data-driven decision-making. Based on this SLR, Table 9 summarizes the limitations of available BDMMs.

**Table 9.** The limitations of existing BDMMs.

| Limitations | Existing BDMMs | | | | | | | | | | | | | | |
|---|---|---|---|---|---|---|---|---|---|---|---|---|---|---|---|
| | A1 | A2 | A3 | A4 | A5 | A6 | A7 | A8 | A9 | A10 | A11 | A12 | A13 | A14 | A15 |
| | [45] | [46] | [47] | [48] | [38] | [49] | [50] | [51] | [52] | [43] | [42] | [53] | [34] | [44] | [54] |
| 1. Poor documentation about the model | | √ | | | | √ | √ | √ | √ | | | | | | |
| 2. No software assessment tool | | √ | √ | √ | | | √ | √ | √ | √ | √ | √ | √ | √ | √ |
| 3. No visualization report | | √ | | √ | √ | √ | √ | √ | √ | √ | √ | √ | √ | √ | |
| 4. No self-assessment tool | √ | √ | √ | √ | √ | √ | √ | √ | √ | √ | √ | √ | √ | √ | |
| 5. Assessment dimensions and sub-dimensions not identified | | √ | | √ | √ | √ | √ | √ | √ | | | √ | | √ | |
| 6. Assessment methods not identified | √ | √ | √ | √ | √ | √ | √ | √ | √ | √ | √ | √ | √ | √ | |
| 7. Limited validation | √ | √ | √ | √ | √ | √ | √ | √ | √ | √ | √ | √ | √ | √ | |
| 8. Poor reliability | √ | √ | √ | √ | √ | √ | √ | √ | √ | | | √ | | | |
| 9. No evaluation in a real case study | √ | √ | √ | √ | √ | √ | √ | √ | √ | | | √ | | | |
| 10. The 5 CMM levels not adapted | | | | √ | √ | √ | √ | √ | √ | | | √ | √ | | √ |
| 11. Sources of assessment components not identified | √ | √ | √ | √ | √ | √ | √ | | √ | √ | √ | √ | √ | √ | |
| 12. Development procedures not identified | √ | √ | √ | √ | √ | √ | √ | √ | √ | | | √ | | | |

## 5. Conclusions

As the global economy gradually recovers from the health crisis due to the COVID-19 pandemic, organizations must redefine their priorities and gain insight from their Big Data. The success of organizations that capitalize on Big Data is due to adopting mature designs before rolling out their implementation. This paper comprised a systematic literature review of the existing Big Data maturity models in the last 15 years (2007–2022) to answer three predefined research questions: RQ1: "What are the existing maturity assessment models for Big Data?", RQ2: "What are the assessment dimensions for Big Data maturity models?", and RQ3: "What are the assessment tools for Big Data maturity models?". A final list of 15 high-quality articles and models was extracted and analyzed to answer the predefined research questions and analyze the existing models' shortcomings. This paper concludes that limited publications from the academic side about available BDMMs need more investigation.

Moreover, this paper presents a basic reference with essential insights for relevant stakeholders to select more-effective assessment models that fit within their organization. In addition, this paper will guide future work to assess and evaluate the existing Big Data maturity as-

sessment models by experts. Future work will provide more details about the assessment dimensions toward developing a new maturity assessment model for Big Data maturity.

**Author Contributions:** Conceptualization, Z.A.A.-S., M.H.H., S.M.S.-M., R.A., R.A.Z., L.A. and A.H.G.; methodology, Z.A.A.-S.; formal analysis, Z.A.A.-S.; writing—original draft preparation, Z.A.A.-S., M.H.H., S.M.S.-M., R.A., R.A.Z., L.A. and A.H.G.; writing—review and editing, Z.A.A.-S., M.H.H., S.M.S.-M., R.A., R.A.Z., L.A. and A.H.G.; visualization, L.A.; supervision, L.A. and A.H.G.; project administration, Z.A.A.-S. All authors have read and agreed to the published version of the manuscript.

**Funding:** This research received no external funding.

**Institutional Review Board Statement:** Not applicable.

**Informed Consent Statement:** Not applicable.

**Data Availability Statement:** Not applicable.

**Conflicts of Interest:** The authors declare no conflict of interest.

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
