# Peer review of "Big Data Maturity Assessment Models: A Systematic Literature Review"

_2504-2289, doi:10.3390/bdcc7010002_

Round 1

Reviewer 1 Report

Dear Authors,

Thank you for the opportunity to review your article. It raises an important issue for the scientific sphere, which is the analysis of methodologies assessing the maturity of big data in specific application contexts. However, I have noticed several issues significantly affecting the quality of your manuscript that I urge you to improve:

1. None of the figures has its source (either external citation or own elaboration).

2. Some of the figures are of unacceptably low quality (eg Figures 4, 8)

3. The article has numerous errors related to punctuation, spelling and formatting.

4. The rigor of the conducted research seems to have gaps which affect the quality of the presented results. For example, when referring to the source [52], it should be borne in mind that this model was developed in 2018 (https://doi.org/10.3390/su10103734 - conceptual foundations described here: https://doi.org/10.1007/978-3-030-40417-8_14), which is not shown in the screening results. In 2020, an article was also published with the proposal maturity model for big data analytics (in the context of airline network planning): https://doi.org/10.1016/j.jairtraman.2019.101721. This year, a preliminary concept of the "global" big data maturity model was proposed, trying to assess the degree of maturity in a more holistic way: https://doi.org/10.1109/ICDS50568.2020.9268720. The innovative "adaptive" big data maturity model was presented in 2022: https://doi.org/10.1016/j.eswa.2022.117965. The screening results give a distorted picture that the subject of big data maturity has not been developed since 2017, where the last research work comes from.

5. In the case of describing the methodology, there was no description of which EBSCOhost and Web of Science databases collections were used.

6. The article unclearly shows how the Big Data Maturity Assessment Dimensions were categorized / classified. On what basis were specific dimensions selected for a given category? The reader should be informed about it in a clear, legible and direct manner. The current table 10 is in my opinion insufficient and the way of its construction should be described in a more extensive manner in the text. At this point, it is worth emphasizing the preceding remark regarding the sources of illustrations, because, for example, the reader does not know whether Figure 12 is own elaboration, or whether these categories of dimensions have already been presented in some published scientific source.

7. Figure 5 is illegible - the percentage distribution should have higher bars. Ideally, percentages should be merged with numeric values so that you don't have to divide the chart into two separate parts. In addition, in the case of this picture, you can also add a trend analysis, as was the case with the literature review in the field of big data analytics for customer insight: https://doi.org/10.3390/app11156993

To sum up, the article is important in terms of the topic discussed and is an extremely important future security for literature sources in the field of big data maturity models. Unfortunately, the article has noticeable methodological errors negatively affecting the quality of the study. Therefore, I suggest applying my suggestions for amendments and thorough revision by yourself, which in my opinion will increase the value of the article presented by you and will be suitable for further publication.

Author Response

Comment:  None of the figures has its source (either external citation or own elaboration).

Response: 

Thank you for the valuable comment.

  • The sources and references of the figures had been updated and highlighted in the attached manuscript file.
  • The elaboration was added to the figures and highlighted in the master file.

The explanation for each phase in Figure 3 had been provided at the beginning of each phase in the paper as highlighted in the attached updated manuscript.

The explanation of  Figure 1:

As shown in Figure 1, the CMM was published in 1993 with five (5) continuous stages of maturity, which are: 1. Initial; 2. Repeatable; 3. Defined; 4. Managed; and 5. Optimized.

  1. Level 1 is “Initial” where processes are not controlled and are unpredictable
  2. Level 2 is “Repeatable” where processes are characterized for Specific organizations but are often reactive.
  3. Level 3 is “Defined” as where processes are standardized and typically documented
  4. Level 4 is “Managed” where processes are measured and controlled
  5. Level 5 is “Optimized”, where processes have a focus on continuous improvement

The explanation of  Figure 2:

The CMM had been modified to be the Capability Maturity Model Integration (CMMI) [31], [32]. Referring to Figure 2, CMMI was published with five (5) continuous stages namely 1-Initial; 2-Managed; 3-Defined; 4-Quantitatively, and 5-Optimized, to get an effective improvement of the organization’s practices and performance [29].

  1. Level 1 is “Initial” where processes are not controlled and are unpredictable
  2. Level 2 is “Managed” where processes exist but are often reactive
  3. Level 3 is “Defined” as where processes are standardized and typically documented
  4. Level 4 is “Quantitatively Managed” where processes are measured and controlled

Level 5 is “Optimized”, where processes have a focus on continuous improvement

Comment:  Some of the figures are of unacceptably low quality (eg Figures 4, 8)

Response: The quality of Figures 4 and 8 had been improved.

Comment:  The article has numerous errors related to punctuation, spelling, and formatting

Response: We checked the language  and improved it for the whole paper

Comment:  

The rigor of the conducted research seems to have gaps which affect the quality of the presented results. For example, when referring to the source [52], it should be borne in mind that this model was developed in 2018 (https://doi.org/10.3390/su10103734 - conceptual foundations described here: https://doi.org/10.1007/978-3-030-40417-8_14), which is not shown in the screening results.

In 2020, an article was also published with the proposal maturity model for big data analytics (in the context of airline network planning): https://doi.org/10.1016/j.jairtraman.2019.101721.

This year, a preliminary concept of the "global" big data maturity model was proposed, trying to assess the degree of maturity in a more holistic way: https://doi.org/10.1109/ICDS50568.2020.9268720. The innovative "adaptive" big data maturity model was presented in 2022: https://doi.org/10.1016/j.eswa.2022.117965. The screening results give a distorted picture that the subject of big data maturity has not been developed since 2017, where the last research work comes from.

Response: 

Thank you for your valuable comments.

Based on your valuable comment, The model by  https://doi.org/10.1016/j.jairtraman.2019.101721  in 2020 had been added to our paper as highlighted in the attached manuscript.

The paper of https://doi.org/10.1007/978-3-030-40417-8_14  published in 2018 by Maria Mach-Król (“Conceptual Foundations for the Temporal Big Data Analytics (TBDA) Implementation Methodology in Organizations”). The study creates conceptual foundations for the temporal big data (TBDA) implementation methodology. The paper presented the most important challenges for big data analytics and the approaches for implementing BDA in organizations, the most important requirements for TBDA implementation methodology, elaborated by the author are pointed out.

After reviewing the paper, the paper focused on Big Data implementation and not proposed a Big Data Maturity Assessment Model. This paper was excluded from our SLR based on our SLR criteria where it did not provide a Big Data Maturity Model and levels.

The study of https://doi.org/10.1109/ICDS50568.2020.9268720, is part of a global objective of creating an end-to-end practical framework for better adoption and implementation of Big Data Technology, with a special focus on North African companies. This paper presents the first major step toward a global Big Data Maturity Model. Moreover, this model was not validated and evaluated. Also, this model did not go through the levels part.

[52] M. Mach-Król, “A Survey and Assessment of Maturity Models for Big Data Adoption,” 11th Int. Conf. Strateg. Manag. Its Support by Inf. Syst., pp. 391–399, 2015. 

Comment:  In the case of describing the methodology, there was no description of which EBSCOhost and Web of Science databases collections were used.

Response:  In this stage, we used the Universiti Sains Malaysia (USM) Database which includes the papers from EBSCOhost and Web of Science.

This SLR excluded the “Web of Science” from the main databases due to the limited availability of the related information. The ResearchGate, Google, Google Scholar, and USM Library Repository were chosen also in order to have a manual search and included sources by the industry that could be found using the search engine Google.

Comment:  The article unclearly shows how the Big Data Maturity Assessment Dimensions were categorized / classified. On what basis were specific dimensions selected for a given category? The reader should be informed about it in a clear, legible and direct manner. The current table 10 is in my opinion insufficient and the way of its construction should be described in a more extensive manner in the text. At this point, it is worth emphasizing the preceding remark regarding the sources of illustrations, because, for example, the reader does not know whether Figure 12 is own elaboration, or whether these categories of dimensions have already been presented in some published scientific source

Response: Thank you so much for your valuable comment.

“Section 4.” A Proposed Classification for the Big Data Maturity Assessment Dimensions had been dropped from the manuscript where it will be in the future work toward developing a Big Data Maturity Model.

Comment:  Figure 5 is illegible - the percentage distribution should have higher bars. Ideally, percentages should be merged with numeric values so that you don't have to divide the chart into two separate parts. In addition, in the case of this picture, you can also add a trend analysis, as was the case with the literature review in the field of big data analytics for customer insight: https://doi.org/10.3390/app11156993

Response: Thank you. Figure 5 had been updated in attached manuscript.

Reviewer 2 Report

Although the paper is thorough and detailed but following are expected while conducting an SLR. 

1. The search queries for the collection of the articles from listed databases. Please try to incorporate that query. This is important for readers to replicate your practice for their work.

2. Please clearly list the future agendas for the other researchers to follow.

Thank you for submitting the article.

Author Response

Comment: The search queries for the collection of the articles from listed databases. Please try to incorporate that query. This is important for readers to replicate your practice for their work.

Response: Thank you for your valuable comment.

The query had been highlighted on (Page 10)

(Big Data OR Big Data Analysis, OR Big Data Maturity) AND (Maturity OR Maturity Assessment OR Maturity) AND (Big Data Maturity Assessment Model OR Big Data Maturity Model OR Big Data Maturity OR Big Data Analytics Maturity Models OR Big Data Capability Maturity Model).

Comment: Please clearly list the future agendas for the other researchers to follow.

Response: Thank you for your comments.

The manuscript updated to highlight future work and highlighted on Page (1) and (30).

Future work will investigate the priority of the Big Data maturity assessment dimensions towards developing a new Big Data Maturity Model. (Page 1)

“This paper will act as a guide for future work to assess and evaluate the existing Big Data Maturity Assessment Models by experts. Future work will provide more details about the assessment dimensions toward developing a new maturity assessment model for big data maturity.” (Page 30)

Round 2

Reviewer 1 Report

Ladies and Gentlemen, thank you for submitting the corrected version of the article. In its present form, it has gained a lot of value and is scientifically sound. Figures now look much better than in the previous version. The authors also did significant work on linguistic and grammatical correctness. I would also like to thank you for your explanations regarding the selection of literature sources for the study. Due to the fact that I did not know what search criteria you used for the literature review (now this has been corrected), I thought it had flaws. I still think that it could have been done better, but in the current form and after your explanation (focusing only on aspects related to the assessment), I am allowing it to be published. Still, only Figure 1 and 2 have the sources from which they come. If the figure is the result of your own work, the caption should include the phrase "Source: own elaboration" or similar. Otherwise, you must provide a source. I am asking for an appropriate correction. Thank you for your cooperation.

Author Response

Dear Reviewer,

Thank you for your valuable comments and the requested enhancements in this paper.

Figures 1 and 2 are from the results of our research and their caption have been modified with "Source: Own Elaboration"